# COVID-19 Vaccines and Atrial Fibrillation: Analysis of the Post-Marketing Pharmacovigilance European Database

**DOI:** 10.3390/biomedicines11061584

**Published:** 2023-05-30

**Authors:** Rosanna Ruggiero, Maria Donniacuo, Annamaria Mascolo, Mario Gaio, Donato Cappetta, Concetta Rafaniello, Giovanni Docimo, Consiglia Riccardi, Imma Izzo, Donatella Ruggiero, Giuseppe Paolisso, Francesco Rossi, Antonella De Angelis, Annalisa Capuano

**Affiliations:** 1Campania Regional Centre for Pharmacovigilance and Pharmacoepidemiology, 80138 Naples, Italy; 2Department of Experimental Medicine, University of Campania L. Vanvitelli, 80138 Naples, Italy; 3Department of Biological and Environmental Sciences and Technologies, University of Salento, 73100 Lecce, Italy; 4Department of Advanced Medical and Surgical Sciences, University of Campania L. Vanvitelli, 80138 Naples, Italy

**Keywords:** atrial fibrillation, adverse events following immunization, COVID-19 vaccines, post-marketing surveillance, EudraVigilance, cardiac safety

## Abstract

Atrial fibrillation (AF) has been described in COVID-19 patients. Recently, some case reports and US pharmacovigilance analyses described AF onset as a rare adverse event following COVID-19 vaccination. The possible correlation is unclear. We systematically analyzed the reports of AF related to COVID-19 vaccines collected in the European pharmacovigilance database, EudraVigilance (EV), from 2020 to November 2022. We carried out descriptive and disproportionality analyses. Moreover, we performed a sensitivity analysis, excluding the reports describing other possible alternative AF causes (pericarditis, myocarditis, COVID-19, or other drugs that may cause/exacerbate AF). Overall, we retrieved 6226 reports, which represented only 0.3% of all those related to COVID-19 vaccines collected in EV during our study period. AF reports mainly referred to adults (in particular, >65 years old), with an equal distribution in sex. Reports were mainly related to tozinameran (54.04%), elasomeran (28.3%), and ChAdOx1-S (14.32%). The reported AF required patient hospitalization in 35% of cases and resulted in a life-threatening condition in 10% of cases. The AF duration (when reported) was highly variable, but the majority of the events had a short duration (moda = 24 h). Although an increased frequency of AF reporting with mRNA vaccines emerges from our study, other investigations are required to investigate the possible correlation between COVID-19 vaccination and the rare AF occurrence.

## 1. Introduction

Coronaviruses are a group of viruses that can affect the airways, leading to severe respiratory disease and fatal outcomes [1]. Since December 2019, the SARS-CoV-2 pandemic has caused high morbidity and mortality worldwide, with over 750 million confirmed cases and over 6.8 million deaths [2]. SARS-CoV-2 infection may remain asymptomatic in the early stages, until the onset of severe pneumonia, dyspnea, organ dysfunction, and even death [3]. Based on the millions of infected cases and the high transmissibility of the virus, researchers have focused their attention on the risk and protective factors of COVID-19. Older age, male sex, and certain ethnicities, as well as pre-existing cardiovascular diseases (hypertension) or pulmonary disease, represent some risk factors for developing severe COVID-19 [4]. In this scenario, COVID-19 vaccines represent the best preventive strategy for protection from disease progression and poor clinical outcomes [5,6]. Various vaccine platforms have been used, including mRNA, non-replicating viral vector, protein subunit, and inactivated-virus-based vaccines [7,8].

As for all medicines, the large-scale use of COVID-19 vaccines has brought to light rarer safety issues, which were unknown at the time of authorization, emerging in the real-world context [9]. Alongside well-known and moderate adverse events, i.e., fever, fatigue, and pain at the site of inoculation, cardiovascular events represent the most serious complications following COVID-19 vaccination. These include myocarditis and pericarditis, mainly associated with mRNA vaccines, while immune thrombocytopenia has been mainly related to viral vector ones [10,11,12]. Another emerging rare cardiac adverse event following anti-COVID-19 immunization is atrial fibrillation (AF). It represents the most common type of heart arrhythmia, whose development is multifactorial and can be related to different comorbidities [13]. AF has also been reported to occur with COVID-19, although its relationship with myocarditis, lung injury, and systemic inflammation status is unclear; interestingly, a study demonstrated increased mortality and prolonged hospitalization in COVID-19 patients who developed AF [14]. Data on the occurrence of AF following COVID-19 vaccination are still limited [15]. The analysis of data from spontaneous reporting systems offers a fundamental tool allowing the identification of new safety signals, which subsequently require further investigations through different methodological approaches [16]. The collection and analysis of adverse events following immunization (AEFIs) has a key role in the monitoring of safety aspects throughout the COVID-19 vaccination campaign, especially in the early stages [17,18]. Recently, an analysis of the reports describing AF after COVID-19 vaccination, which were collected in the US Vaccine Adverse Event Reporting System (VAERS), has been published [19]. After more than 500 million doses of COVID-19 vaccine administered, approximately 2500 AF events were recorded (5 per million doses), equally distributed between male and female individuals and between the first and second dose of the vaccine; the majority of patients were ≥40 years old [19]. Another recent real-world pharmacovigilance study based on VAERS evaluated the incidence of AF in association with vaccines, including the COVID-19 ones [20]. The study showed that COVID-19 vaccines were associated with a higher occurrence of AF events compared to other vaccines (herpes zoster, influenza, pneumococcal vaccines) [20]. AF represents a rare adverse event reported in a very small number of vaccinated patients. To our knowledge, no analyses of European data are yet available. Therefore, the present study aimed at evaluating the occurrence of AF reported as an AEFI following COVID-19 vaccination by analyzing the European post-marketing database EudraVigilance (EV), since the analysis of international databases allows the extrapolation of important safety information and signals [21].

## 2. Materials and Methods

### 2.1. Study Design

This is a retrospective observational study carried out on data available from the reports of AEFIs related to COVID-19 vaccines sent in Europe to the regulatory medicine agencies from 1st December 2020 to 28th November 2022. We retrieved all reports describing AF cases that occurred in subjects vaccinated against COVID-19 and collected in the European post-marketing database EV. For our study, we included the mRNA-based vaccines “elasomeran” (Moderna) and “tozinameran” (Pfizer-BioNTech); the viral vector vaccines, “AD26.COV2.S” (Janssen) and “ChAdOx1-S” (AstraZeneca); and the recombinant subunit protein vaccine “NVX-CoV2373” (Novavax).

### 2.2. AEFI Data Collection

According to the European pharmacovigilance rules, when an adverse event is identified by a patient or healthcare professional and there is a suspicion of its possible association with a vaccine (such as the COVID-19 one) or a pharmacological treatment, reporting to the national or international drug competent authorities has to be carried out. All national reports are compiled in a unique European pharmacovigilance database, named EudraVigilance. The EV data are publicly available through the European Medicines Agency (EMA) website (www.adrreports.eu, accessed on 28 November 2022).

### 2.3. Descriptive Analysis

From each report, we retrieved the following information: patient data (age group and sex); report data, including the reporting type (spontaneous or emerging from a study), receipt date, primary source qualification (non-healthcare or healthcare professional (non-HCP or HCP)), and the primary source country (European or Non-European Economic Area); COVID-19 vaccination data, including the type of suspected vaccine (mRNA-based, viral vector-based, or recombinant subunit protein), the vaccination type (heterologous or homologous), and the number of administrated doses; AF data, in terms of duration (when available), outcome, and seriousness criteria; all AEFIs that occurred after vaccination overlapping with the AF (categorized according to the Medical Dictionary for Regulatory Activities (MedDRA) in the reference “System Organ Classes” (SOCs)); and all other suspected or concomitant drugs described in the reports for each included vaccine.

We investigated the other overlapping AEFIs, to identify which combination of other adverse events was more frequently reported. Moreover, we examined the other suspected and concomitant drugs that were reported, in order to identify the patients with pre-existing diseases or disturbances that could represent risk factors for the occurrence of AF. We analyzed other drugs that may cause/exacerbate AF or atrial flutter [22], reported as other suspected or concomitant drugs in individual case safety reports (ICSRs).

### 2.4. Disproportionality Analysis

To compare the frequency of AF reporting for each COVID-19 vaccine, we calculated the reporting odds ratio (ROR) 95% CI. The ROR was computed by comparing each COVID-19 vaccine, using mRNA vaccines as the main reference, considering their already known cardiac safety aspects. The RORs were computed on ICSRs since these are publicly available on the EMA website. The signal was considered statistically significant when the lower limit of the 95% CI of an ROR exceeded 1.0.

Moreover, we carried out a sensitivity analysis excluding ICSRs that reported “pericarditis”, “myocarditis”, or “COVID-19 infection” as other adverse events, considering them as other possible causes of AF, as well as ICSRs reporting other drugs that may cause/exacerbate AF [22]. Forest plots were obtained for both comparisons using R (Version 4.1.2 (1 November 2021); R Development Core Team).

## 3. Results

### 3.1. Descriptive Analysis

Overall, we retrieved from EV 6226 ICSRs describing AF as an adverse event and at least a COVID-19 vaccine as a suspected drug. As reported in Table 1, the AF ICSRs represented 0.3% of all ICSRs related to COVID-19 collected during our study period.

Excluding duplicates, an overall total of 6160 ICSRs emerged (Table 2). In particular, ICSRs were mainly related to tozinameran (N = 3330; 54.04%), followed by elasomeran (N = 1743; 28.3%) and ChAdOx1-S (N = 882; 14.32%). Demographic characteristics of ICSRs are described in Table 2. Overall, the main age groups were 65–85 and 18–64 years, emerging in 47.45% (N = 2923) and in 39.46% (N = 2431) of all ICSRs, respectively. This distribution persisted for all single vaccines, regardless of the vaccine type. Only for AD26.COV2.S-related ICSRs did we observe an inverted distribution between these two age groups, with a major distribution of these ICSRs in the 18–64 age group (46.53%), followed by the 65–85 age group (38.2%). Overall, pediatric age groups were underrepresented (N = 12). In particular, two of these ICSRs referred to a patient of 0–1 months and another of 3–11 years, vaccinated with tozinameran, while only one ICSR was related to a patient aged between 2 months and 2 years who had been vaccinated with elasomeran. The remaining pediatric ICSRs referred to adolescents (12–17 age group) mainly vaccinated with tozinameran (N = 8) or ChAdOx1-S (N = 1). Only 59 reports (<1%) were categorized as related to a mixed vaccination, since they reported different COVID-19 vaccines as suspected drugs. Considering the number of COVID-19 vaccine doses, the majority of ICSRs referred to a single vaccination dose (more than 98% for both all vaccines and each single examined vaccine). The few ICSRs that referred to second (N = 85) or third (N = 9) doses were mainly represented by heterologous vaccination, since, in 52 out of 85 ICSRs with two doses and in 7 out of 9 ICSRs, three vaccine doses were reported for different COVID-19 vaccines as suspected drugs. Sex was equally distributed, with minimal percentage differences except for ChAdOx1-S (M = 58.33%; F = 38.88%) and mixed vaccination (M = 61.02%; F = 37.29%). Overall, the majority of ICSRs were equally distributed in terms of the primary source (healthcare personnel = 54.77%; non-healthcare personnel = 45.23%), as well as for the country of reporting (European Economic Area = 51.71%; non-European Economic Area = 48.29%). However, a major contribution of healthcare personnel as reporters (72.86%) emerged only for elasomeran ICSRs, while the distribution of the countries of reporting varied when considering the single vaccine. Non-European Economic Areas were more represented as a country source for elasomeran (71.2%) and AD26.COV2.S (65.97%) ICSRs.

Regarding the clinical characteristics of AF events, from our analysis, it emerged that the median event duration, reported in 1018 out of 6160 ICSRs, was 48 h (IQR:312). The reported duration of this adverse event was highly variable, from 3 min up to 489 days. The majority of the events had a short duration (Figure 1). The duration value most frequently reported was 24 h.

The distribution of all AF events’ outcomes and seriousness criteria for each COVID-19 vaccine is described in Table 3. When AF was considered by the reporter as a serious event, in the majority of the cases (39%), it was categorized as a medically important event (as per seriousness criteria). Moreover, AF required patient hospitalization in 35% of cases and resulted in a life-threatening condition in 10% of cases. In addition, 2% of the reported AF events resulted in disability for the patient.

Regardless of the reported suspected COVID-19 vaccine, the event outcome was favorable in 40% of cases, resulting in the complete resolution (26.2%) or improvement (13.7%) of the event. On the other hand, the AF outcome was unfavorable in 38% of cases, resulting in a resolved with sequelae, not recovered, or fatal condition in 4.7%, 30.4%, and 3% of ICSRs, respectively.

AF that occurred following COVID-19 vaccination was frequently reported with other overlapping adverse events. As reported in Table 4, these events mainly belonged to general disorders and administration site conditions, among which fatigue, pyrexia, chest pain, asthenia, and malaise were the most frequently reported. Other cardiac disorders that often overlapped with AF included palpitations, arrhythmia, tachycardia, cardiac failure, pericarditis, and myocarditis. Moreover, the onset of COVID-19 also emerged among the first twenty other adverse events overlapping with AF (Table 5).

Suspected drugs other than vaccines were reported in a few ICSRs. The majority of ICSRs (>90%), even when analyzed for each single vaccine, did not report other suspected drugs. Moreover, 2257 ICSRs reported one or more concomitant drugs (Table 1). In 199 ICSRs, drugs used for AF treatment were among concomitants. These mainly included direct-acting oral anticoagulants (DOACs, e.g., apixaban, rivaroxaban, edoxaban), beta-blockers (metoprolol, bisoprolol, carvedilol, atenolol, and sotalol), and arrhythmic drugs, such as flecainide and amiodarone. We carried out an analysis of other drugs that may cause/exacerbate AF reported as other suspected or concomitant drugs (Table 6). We found several ICSRs that reported antiarrhythmic drugs, mainly flecainide, as concomitant drugs. A few ICSRs reported other drugs that may cause/exacerbate AF as suspected drugs. These latter were mainly represented by amiodarone, ibrutinib, 5-fluorouracil, epinephrine, and caffeine.

### 3.2. Disproportionality Analysis

According to the RORs computed on the original dataset, significant results for all considered comparisons emerged. In particular, we found that the mRNA vaccines were associated with an increased reporting probability of AF (Figure 2). Moreover, elasomeran was associated with a higher reporting frequency compared to tozinameran (ROR = 1.71; CI 95% 1.61–1.81), while comparing the two viral vector vaccines showed an ROR that was statistically significant for AD26.CoV2.S compared to ChAdOx1-S (ROR = 1.25; CI 95% 1.05–1.50). NVX-CoV2373 was excluded from the disproportionality analysis since only two ICSRs were reported.

Moreover, we conducted a sensitivity analysis excluding 541 ICSRs in which pericarditis (N = 192), myocarditis (N = 151), and/or COVID-19 (N = 241) were reported as other adverse events. In addition, we excluded all ICSRs in which drugs causing or exacerbating AF were reported (Table 6). Excluding ICSRs reporting the above-mentioned confounding factors, 5324 ICSRs emerged. On this dataset, a sensitivity analysis was carried out, applying new RORs (Figure 3). Albeit slightly lower, signals remained statistically significant, except for the comparison between the two viral vector vaccines (ROR= 1.09; 95% CI 0.89–1.33).

## 4. Discussion

To our knowledge, this is the first analysis of pharmacovigilance European data on AF following COVID-19 vaccination. The current study aimed to evaluate the reporting frequency of AF related to COVID-19 vaccines by using data retrieved from the European pharmacovigilance database EV.

Overall, we found and analyzed more than 6000 reports describing AF as an AEFI with anti-COVID-19 vaccines collected in the EV database. These European AF reports were more than twice as frequent as those that emerged from a recent pharmacovigilance study conducted on data collected in the US VAERS database during an almost overlapping study period [19]. In particular, from 2020 to 2022, a total of 2611 AF events that occurred after COVID-19 vaccination were collected in the VAERS database. Out of these AF cases, 315 were new-onset events.

The AF reports included in our dataset represented a very small percentage of all ICSRs related to COVID-19 vaccines sent to the European regulatory agencies during our study period. Moreover, from other analyses, AF resulted in a very rare complication after vaccination. Even if major compared to subjects not vaccinated against COVID-19, the incidence of cardiac arrhythmia following COVID-19 vaccination is very low [23].

As expected, the majority of the retrieved reports were related to mRNA vaccines, especially to tozinameran. This can be related to the wider use of mRNA vaccines compared to the other platform vaccines. In line with the extensive vaccination campaign aimed primarily at adults, the majority of the reports emerging from our dataset referred to the age groups of 65–85 years and 18–64 years.

Interestingly, our results showed an important percentage of reports sent by non-HCP. These ICSRs were almost equivalent to those sent by HCP. Even if citizen involvement in the ADR reporting could be associated with the lower quality or reliability of data, they represent an important information source. In the last few years, all drug regulatory agencies have aimed to implement their full involvement also in this aspect. This is in line with the centrality and the empowerment of patients in the care process, which they are aiming for.

This could highlight the important impact that COVID-19 vaccine development and use has had on the general interest regarding the safety aspects of such vaccines. The innovative aspects of these vaccines, both in terms of the used platforms and in terms of development and authorization timing, have brought about pharmacovigilance activities, to which HCPs and others can and should contribute. However, the enormous media coverage and the following high vaccination hesitancy indicate the necessity of better and correct dissemination regarding the sense and meaning of these activities. This requires an understanding of pharmacovigilance as a cultural aspect to be implemented also at a qualitative level. This could allow us to improve the data’s quality and reliability, making them more reliable even when they come from a citizen, as in our dataset.

In contrast with VAERS analysis, from which an almost equal number of AF followed the first as well as the second dose [19], the majority of our AF reports mainly referred to a single-dose COVID-19 vaccine. This could be related to the underreporting phenomenon or the low quality of data, as bias can characterize the spontaneous reporting system. Moreover, it could be associated with the major susceptibility of patients to AF that emerges after the first dose, but, to our knowledge, there is no clear plausibility in these data. However, our results are in line with an increased risk of AF or flutter arrhythmia following the first dose of the mRNA-1273 vaccine, which emerged from a subgroup analysis of a study conducted in patients with cardiac implantable electronic devices to ascertain the relationship between vaccination and arrhythmic events [15].

As is well known, during the vaccinal campaign, cardiac safety aspects of COVID-19 vaccines, especially those that are mRNA-based, emerged [24,25,26,27]. The attention of regulatory agencies, as well as the scientific and general community, has been mainly focused on the adverse events of myocarditis and pericarditis. These ones have emerged as rare complications in young vaccinated males [11,27]. Instead, unlike what emerged for major cardiac adverse events related to the COVID-19 vaccine (myocarditis and pericarditis) with a higher incidence in men, the reports of interest to us were equally distributed between the two sexes. In contrast with our results, in an online cohort study, the female sex was identified as one of the risk factors most strongly associated with the occurrence of serious AEFIs with COVID-19 vaccines [28]. Thus, data regarding the sex influence on ADR susceptibility are conflicting.

Recently, attention has also shifted to other, even rarer cardiac adverse events, such as AF following mRNA vaccination [15,19,23,29]. Considering the important impact of this arrhythmia in terms of hospitalization, morbidity, and mortality [30], data analysis and insights into the possible relationship between AF and COVID-19 vaccination represent an important research topic.

As is well known, AF is an age-related disease where comorbidities, such as hypertension, diabetes mellitus, obesity, chronic kidney disease, and inflammatory diseases, play a pivotal role [31]. Functional changes such as altered intracellular Ca2+ levels appear to be responsible for short-term remodeling, while structural remodeling, particularly atrial fibrosis, resulting from the high atrial rate, is a major feature of AF recurrence and is largely irreversible [32,33]. Furthermore, research is highlighting emerging aspects of the homeostatic mechanisms affecting atrial remodeling in AF [30].

To date, AF is considered a serious form of arrhythmia that can occur in COVID-19 patients. New-onset AF represents a frequent complication of COVID-19 [34]. Several published works support the possible cardiac damage related to SARS-CoV-2 infection [35,36]. In particular, mortality related to AF increased during the COVID-19 emergency in the United States [37]. Aberrant inflammatory responses, as well as activation and/or dysregulation of the immune system, are involved in the biological plausibility of this COVID-19 complication [38].

The possible underlying mechanism of AF following COVID-19 vaccination remains unclear. Local or systemic inflammation, including cardiac inflammation of the endothelium, induced by COVID-19 vaccination has been proposed as a hypothetical mechanism on the basis of AF after COVID-19 immunization [29], but it is difficult to establish a clear correlation only based on case reports or pharmacovigilance data.

Analyzing the possible relationship between AF and COVID-19 vaccination, one should consider some possible confounding factors, including concomitant and pre-existing diseases and concomitant therapies, as well as COVID-19 itself.

AF has been considered a possible precursor of mRNA vaccine-induced pericarditis [39]. Some case reports describe COVID-19-associated myocarditis presenting as new-onset AF [40]. For these reasons, we chose to conduct an analysis of sensitivity excluding all the reports in which at least one of the above-mentioned alternative causes of AF was reported.

From our disproportionality analysis, an increased probability of AF reporting emerged for mRNA vaccines compared to viral vector ones. Moreover, when we compared the two mRNA vaccines, elasomeran was associated with an increased frequency of AF risk reporting compared to tozinameran. These results persisted also when ICSRs reporting confounding factors for AF occurrence were excluded in the sensitivity analysis, but only for mRNA vaccines. When we compared the two viral vector vaccines AD26.COV2.S and ChAdOx1-S, the resulting ROR was not statistically significant. A possible association between AF and COVID-19 vaccines emerged also from a recent study conducted on the VAERS database, according to which, among all vaccines, COVID-19 ones were significantly associated with AF [20].

To date, AF is not included in any COVID-19 vaccines’ Summary of Product Characteristics or Risk Management Plan. No signals have emerged in this regard from medicine regulatory agencies. Although we found almost 6000 reports describing AF as a suspected adverse event, it is important to highlight that it could represent a very rare event for which the possible relationship with COVID-19 remains unclear. Some other authors have hypothesized that mRNA vaccination may lead to arrhythmia [29]. According to a recent literature review on the topic, overall, the authors found only six published cases of AF following mRNA COVID-19 vaccination. Moreover, it is important to consider that in four of these cases, the patients presented comorbidities, which could be a predisposing factor for AF, including diabetes, Marfan syndrome, or persistent AF [29].

Our study possesses all the well-known limitations of pharmacovigilance studies. These include the possible underreporting of adverse events, the possible low quality of data, or missing information. At the same time, data from spontaneous reporting systems lack information about some other risk factors (e.g., physical activity, obesity, diabetes), which can be retrieved in ad hoc studies. However, pharmacovigilance studies, especially when based on large datasets retrieved from international databases (such as EV), allow for the identification of potential safety signals. These results require further insights through studies with stronger methodological designs. Our results may contribute to the monitoring of rarer adverse events and they represent a starting point for further investigations, if deemed necessary by regulatory agencies or by the drug industry itself.

## 5. Conclusions

The current study evaluated the reports collected in EV that described AF as an AEFI that occurred following COVID-19 vaccination. Overall, we found and analyzed 6160 ICSRs. Considering the high number of vaccinated patients, the described AF cases represent a rare adverse event reported by a very small part of vaccinated patients. The majority of AF reports were related to mRNA vaccines. From our disproportionality analysis, a higher AF reporting frequency emerged for mRNA vaccines. This was confirmed by our sensitivity analysis, for which we excluded the ICSRs reporting confounding factors, such as COVID-19, myocarditis, pericarditis, and other drugs, causing or exacerbating AF. The possible mechanism of AF following COVID-19 vaccination is unclear. Although an increased frequency of AF reporting with mRNA vaccines emerged from our study, further investigations are required to investigate the possible correlation between COVID-19 vaccination and the rare occurrence of AF.

## Figures and Tables

**Figure 1 biomedicines-11-01584-f001:**
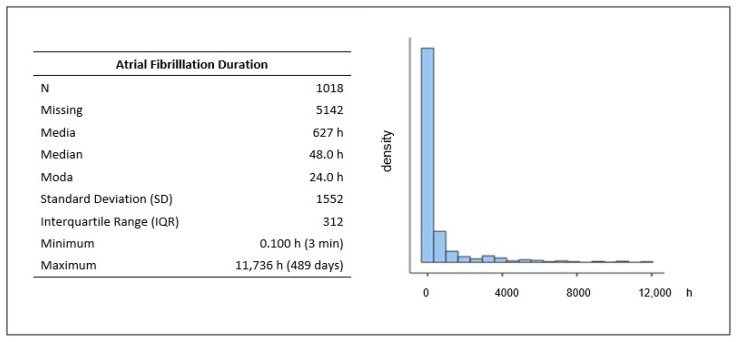
Distribution of atrial fibrillation duration following COVID-19 vaccination and listed as suspected adverse event in EudraVigilance database.

**Figure 2 biomedicines-11-01584-f002:**
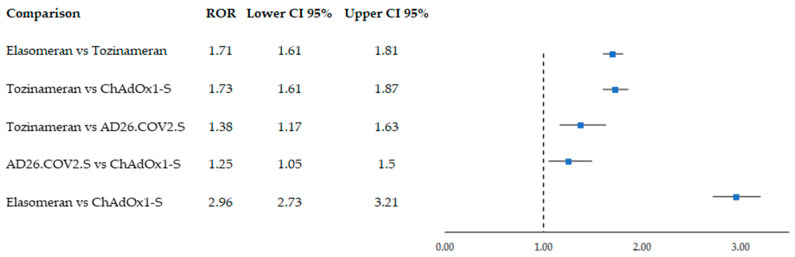
Reporting odds ratios (RORs) obtained on individual case safety reports collected in EudraVigilance from December 2020 to November 2022 and reporting atrial fibrillation following COVID-19 vaccination. Vaccines included were the mRNA-based vaccines elasomeran (Moderna) and tozinameran (Pfizer-BioNTech) and the viral vector vaccines AD26.COV2.S (Janssen) and ChAdOx1-S (Astrazeneca).

**Figure 3 biomedicines-11-01584-f003:**
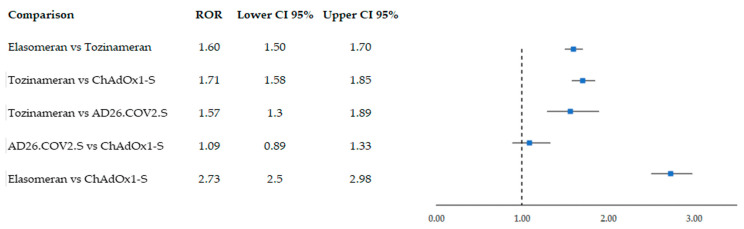
Sensitivity analysis applying reporting odds ratio (ROR) on individual case safety reports collected in EudraVigilance from December 2020 to November 2022 and reporting atrial fibrillation following COVID-19 vaccination. Vaccines included were the mRNA-based vaccines elasomeran (Moderna) and tozinameran (Pfizer-BioNTech) and the viral vector vaccines AD26.COV2.S (Janssen) and ChAdOx1-S (Astrazeneca). For this sensitivity analysis, ICSRs reporting confounding factors for AF occurrence were excluded.

**Table 1 biomedicines-11-01584-t001:** Individual case safety reports (ICSRs), including duplicates, reporting at least an anti-COVID-19 vaccine as a suspected drug and atrial fibrillation as an adverse event following immunization (AEFI) and collected in the European pharmacovigilance database EudraVigilance (EV) up to November 2022.

Anti-COVID-19 Vaccine	Total ICSR in EV	ICSR with AF in EV	ICSR with AF/ Total ICSR
Ad26.Cov2.S (Janssen)	70,111	149	0.2
ChAdOx1-S NCoV-19 (Astrazeneca)	538,202	910	0.2
NVX-CoV2373 (Novavax)	1481	2	0.1
Elasomeran (Moderna)	360,626	1785	0.5
Tozinameran (Pfizer)	1,174,613	3380	0.3
Total	2,074,922	6226	0.3

**Table 2 biomedicines-11-01584-t002:** Demographic characteristics of individual case safety reports (ICSRs) (excluding duplicates) collected from December 2020 to November 2022, and reporting atrial fibrillation as adverse event following COVID-19 vaccination. ICSRs were retrieved from the European spontaneous reporting system EudraVigilance. Vaccines included were the mRNA-based vaccines elasomeran (Moderna) and tozinameran (Pfizer-BioNTech), the viral vector vaccines AD26.COV2.S (Janssen) and ChAdOx1-S (Astrazeneca), and the recombinant subunit protein vaccine NVX-CoV2373 (Novavax).

Variable	Level	All ICSRs (N = 6160; 100%)	Tozinameran ICSRs (N = 3330; 54.04%)	Elasomeran ICSRs (N = 1743; 28.3%)	ChAdOx1-S ICSRs (N = 882; 14.32%)	AD26.COV2.S ICSRs (N = 144; 2.34%)	NVX-CoV2373 ICSRs (N = 2; 100%)	Mixed Vaccination (N = 59; 0.96%)
Age Group	0–1 month	1 (0.02%)	1 (0.03%)	0	0	0	0	0
	2 months–2 years	1 (0.02%)	0	1 (0.06%)	0	0	0	0
	3–11 years	1 (0.02%)	1 (0.03%)	0	0	0	0	0
	12–17 years	9 (0.14%)	8 (0.24%)	0	1 (0.11%)	0	0	
	18–64 years	2431 (39.46%)	1314 (39.47%)	638 (36.60%)	388 (44%)	67 (46.53%)	2 (100%)	22 (37.29%)
	65–85 years	2923 (47.45%)	1501 (45.08%)	920 (52.8%)	414 (47%)	55 (38.2%)	0	33 (55.93%)
	More than 85 years	496 (8.05%)	328 (3.85%)	131 (7.52%)	22 (2.5%)	13 (9.03%)	0	2 (3.39%)
	Missing	298 (4.84%)	177 (5.32%)	53 (3.04%)	57 (6.5%)	9 (6.25%)	0	2 (3.39%)
Sex	F (%)	2957 (48.00%)	1654 (49.68%)	804 (46.13%)	421 (47.7%)	56 (38.88%)	0	22 (37.29%)
	M (%)	3115 (50.57%)	1627 (48.87%)	923 (52.95%)	443 (50.2%)	84 (58.33%)	2 (100%)	36 (61.02%)
	Missing (%)	88 (1.43%)	49 (1.47%)	16 (0.92%)	18 (2.04%)	4 (2.78%)	0	1 (1.69%)
Primary Source	Healthcare Professional	3374 (54.77%)	1566 (47.04%)	1270 (72.86%)	423 (48%)	83 (57.64%)	1 (50%)	31 (52.54%)
	Non-Healthcare Professional	2786 (45.23%)	1764 (52.9%)	473 (27.14%)	459 (52.04%)	61 (42.36%)	1 (50%)	28 (47.46%)
Primary Source Country for Regulatory Purposes	European Economic Area	3185 (51.71%)	2137 (64.19%)	502 (28.8%)	462 (52.4%)	49 (34.03%)	2 (100%)	33 (55.93%)
	Non-European Economic Area	2975 (48.29%)	1193 (35.83%)	1241 (71.2%)	420 (47.6%)	95 (65.97%)	0	26 (44.07%)
Doses of COVID-19 Vaccine	1	6066 (98.47%)	3305 (99.27%)	1742 (99.94%)	873 (98.98%)	144 (100%)	2 (100%)	0
	2	85 (1.38%)	23 (0.69%)	1 (0.06%)	9 (1.02%)	0	0	52 (88.13%)
	3	9 (0.15%)	2 (0.06%)	0	0	0	0	7 (11.86%)
Other Suspected Drugs than COVID-19 Vaccine	0	6039 (98.91%)	3271 (98.26%)	1710 (98.11%)	862 (97.7%)	140 (97.22%)	2 (100%)	54 (91.52%)
	1	94 (1.53%)	44 (1.32%)	27 (1.55%)	16 (1.81%)	3 (2.08%)	0	4 (6.78%)
	2	19 (0.31%)	10 (0.3%)	3 (0.17%)	4 (0.45%)	1 (0.69%)	0	1 (1.69%)
	3	1 (0.02%)	1 (0.03%)	0	0	0	0	0
	4	2 (0.03%)	1 (0.03%)	1 (0.06%)	0	0	0	0
	≥5	5 (0.08%)	3 (0.09%)	2 (0.11%)	0	0	0	0
Concomi-tant Drug(s)	0	3903 (63.36%)	2329 (69.96%)	922 (52.9%)	531 (60.20%)	85 (59.03%)	2	34 (57.63%)
	1	606 (9.84%)	278 (8.35%)	190 (10.9%)	116 (13.15%)	10 (6.94%)	0	12 (20.34%)
	2	348 (5.65%)	165 (4.96%)	106 (6.08%)	68 (7.71%)	9 (6.25%)	0	0
	3	299 (4.85%)	147 (4.41%)	93 (5.33%)	52 (5.89%)	3 (2.08%)	0	4 (6.78%)
	4	223 (3.62%)	108 (3.24%)	78 (4.48%)	28 (31.8%)	9 (6.25%)	0	0
	≥5	781 (12.68)	303 (9.10%)	354 (20.31%)	87 (9.86%)	28 (19.44%)	0	9 (15.25%)

**Table 3 biomedicines-11-01584-t003:** Distribution of outcomes and seriousness criteria of atrial fibrillation reported as adverse event following COVID-19 vaccination and collected from December 2020 to November 2022, in the European spontaneous reporting system EudraVigilance. Vaccines included were the mRNA-based vaccines elasomeran (Moderna) and tozinameran (Pfizer-BioNTech), the viral vector vaccines AD26.COV2.S (Janssen) and ChAdOx1-S (Astrazeneca), and the recombinant subunit protein vaccine NVX-CoV2373 (Novavax).

	Outcome						Seriousness Criteria				
COVID-19 Vaccine	Recovered	Recovered with Sequelae	Recovering	Not Recovered	Fatal	Unknown	Other Medically Important Events	Caused/Prolonged Hospitalization	Resulting in Death	Disabling	Life-Threating
Tozinameran	833 (13.5)	175 (2.8)	521 (2.8)	1017 (16.5)	97 (1.6)	687 (11.2)	1336 (21.7)	1172 (19.0)	101 (1.6)	81 (1.3)	321 (5.2)
Elasomeran	496 (8.1)	50 (0.8)	150 (2.4)	539 (8.8)	60 (1.0)	448 (7.3)	689 (11.2)	646 (10.5)	60 (1.0)	26 (0.4)	202 (3.3)
ChAdOx1-S	241 (3.9)	55 (0.9)	150 (2.4)	249 (4.0)	15 (0.2)	172 (2.8)	310 (5.0)	269 (4.4)	20 (0.3)	27 (0.4)	104 (1.7)
AD26.COV2.S	31 (0.5)	3 (<0.1)	7(0.1)	51 (0.8)	17 (0.3)	35 (0.5)	40 (0.6)	48 (0.8)	19 (0.3)	1 (<0.1)	28 (0.5)
NVX-CoV2373	1 (<0.1)	0	0	0	0	1 (<0.1)	0	2 (<0.1)	0	0	0
Mixed Vaccination	9 (0.1)	7 (0.1)	14 (0.2)	19 (0.3)	1 (<0.1)	9 (0.1)	17 (0.3)	31 (0.5)	1 (<0.1)	4 (0.1)	3 (<0.1)
Total	1611 (26.2)	290 (4.7)	842 (13.7)	1875 (30.4)	190 (3.01)	1351 (21.9)	2392 (38.8)	2168 (35.2)	201 (3.3)	139 (2.3)	658 (10.7)

% of the total reported atrial fibrillation events (N = 6160).

**Table 4 biomedicines-11-01584-t004:** Distribution by system organ class of other adverse events overlapping with atrial fibrillation following COVID-19 vaccination, reported and collected in the EudraVigilance database from December 2020 to November 2022.

System Organ Class	%
General disorders and administration site conditions	20.14
Cardiac disorders	16.18
Nervous system disorders	12.77
Respiratory, thoracic, and mediastinal disorders	9.90
Investigations	9.34
Musculoskeletal and connective tissue disorders	4.55
Gastrointestinal disorders	4.45
Infections and infestations	3.60
Vascular disorders	3.40
Surgical and medical procedures	2.25
Psychiatric disorders	2.14
Skin and subcutaneous tissue disorders	2.14
Injury, poisoning, and procedural complications	1.89
Metabolism and nutrition disorders	1.58
Renal and urinary disorders	1.11
Blood and lymphatic system disorders	1.01
Eye disorders	1.00
Ear and labyrinth disorders	0.54
Immune system disorders	0.47
Social circumstances	0.40
Endocrine disorders	0.31
Hepatobiliary disorders	0.29
Reproductive system and breast disorders	0.26
Neoplasms benign, malignant, and unspecified (incl. cysts and polyps)	0.19
Congenital, familial, and genetic disorders	0.08
Product issues	0.02
Pregnancy, puerperium, and perinatal conditions	0.00

**Table 5 biomedicines-11-01584-t005:** The first twenty other adverse events overlapping with atrial fibrillation following COVID-19 vaccination, reported and collected in EudraVigilance database from December 2020 to November 2022.

Lowest Level Terms	N	%
Dyspnea	901	4.03
Fatigue	695	3.11
Palpitations	613	2.74
Pyrexia	497	2.22
Headache	475	2.12
Arrhythmia	443	1.98
Dizziness	436	1.95
Tachycardia	414	1.85
Chest pain	377	1.69
Asthenia	356	1.59
Malaise	330	1.48
Nausea	284	1.27
Condition aggravated	259	1.16
Cardiac failure	258	1.15
Heart rate increased	252	1.13
Chills	251	1.12
Chest discomfort	221	0.99
Myalgia	222	0.99
Hypertension	218	0.97
COVID-19	171	0.76

**Table 6 biomedicines-11-01584-t006:** Analysis of the concomitant drugs that may cause/exacerbate atrial fibrillation or atrial flutter that were reported as other suspected or concomitant drugs in individual case safety reports (ICSRs) describing atrial fibrillation as an adverse event following COVID-19 vaccination and collected in EudraVigilance during the study period.

Drug Class	Drug	ICSR Reporting Drug as Other Suspected Drug	ICSR Reporting Drug as Concomitant Drug
Antiarrhythmic	Adenosine	0	1
	Amiodarone	2	38
	Flecainide	0	110
	Propafenone	0	22
Tyrosine kinase inhibitor	Ibrutinib	2	5
Antimetabolite	5-Fluorouracil	1	1
Antidepressant (SSRI)	Fluoxetine	0	19
Antiemetic	Ondansetron	0	6
Anti-inflammatory	Diclofenac	0	12
	Etoricoxib	0	6
	Methylprednisolone	0	6
Antiplatelet	Ticagrelor	0	3
Antipsychotic	Clozapine	0	2
	Prochlorperazine	0	3
	Olanzapine	0	2
	Risperidone	0	1
	Quetiapine	0	11
Bisphosphonate	Alendronate	0	13
	Zoledronic acid	0	2
Bronchodilator	Terbutaline	0	1
	Theophylline	0	2
	Ipratropium	0	16
	Tiotropium	0	19
Cannabinoid	Cannabis	0	3
Catecholaminergic	Epinephrine	1	2
Central nervous system Depressant	Alcohol	0	6
I_f_ current inhibitor	Ivabradine	0	5
Immune-modulating agent	Fingolimod	0	1
Opioid	Morphine	0	8
Phosphodiesterase inhibitor	Sildenafil	1	9
	Vardenafil	0	1
Stimulant	Caffeine	1	9

## Data Availability

The data are publicly available through the European Medicines Agency (EMA) website (www.adreports.eu, accessed on 28 November 2022).

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
