# Peer review of "COVID-19 Vaccines and Atrial Fibrillation: Analysis of the Post-Marketing Pharmacovigilance European Database"

_biomedicines, 2023, doi:10.3390/biomedicines11061584_

Round 1

Reviewer 1 Report

Dear editor,

I did review the manuscript detailed below submitted to biomedicine.

COVID-19 vaccines and atrial fibrillation: analysis of the post-marketing pharmacovigilance European database.

The authors analyzed reports of AF related to COVID-19 vaccines collected in the European pharmacovigilance database, Eudra-16 vigilance (EV), from 2020 up to November 2022. 

The analysis is descriptive, but it contains important information. Although the study do answer the question of a causative relation between vaccination and the occurrence of AF, the knowledge about baseline data in these patients. For this reason, the paper is excellent.

The manuscript is well written and the results are carefully presented. Also the discussion is well balanced. I have only two minor points:

1.  Please explain all abbreviation properly.

2. Discuss more the limitation, the reports are mostly sent by non-health care professionals.  

I recommend the publishing this manuscript.

Author Response

Reply to Review 1 Report Form

Dear editor,

I did review the manuscript detailed below submitted to biomedicine.

COVID-19 vaccines and atrial fibrillation: analysis of the post-marketing pharmacovigilance European database.

The authors analyzed reports of AF related to COVID-19 vaccines collected in the European pharmacovigilance database, Eudra-16 vigilance (EV), from 2020 up to November 2022. 

The analysis is descriptive, but it contains important information. Although the study do answer the question of a causative relation between vaccination and the occurrence of AF, the knowledge about baseline data in these patients. For this reason, the paper is excellent.

The manuscript is well-written and the results are carefully presented. Also the discussion is well balanced. I have only two minor points:

  1. Please explain all abbreviation properly.
    Thank you. We revisioned all the abbreviations, adding the missing parts.
  2. Discuss more the limitation, the reports are mostly sent by non-health care professionals. 
    We implemented the discussion section. Please see lines 328-333 and 342-343. 

I recommend the publishing this manuscript.

Reviewer 2 Report

Dear authors,

I have now completed the review of the manuscript titled "COVID-19 vaccines and atrial fibrillation: analysis of the post-marketing pharmacovigilance European database."

In the present study, the authors systematically analyzed the reports of AF related to COVID-19 vaccines collected in the European pharmacovigilance database (Eudravigilance).

The manuscript is interesting and, in general, fair written.

I have some suggestions to further improve the quality of the manuscript.

1. The introduction section introduced some relevant articles. Please explain the results or summarize with effect sizes. 

2. Authors selected Eudravigilance. What is the benefit of the Eudravigilance compared to most well-known pharmacovigilance dataset (WHO Vigibase)? Why authors choose it?

3. What is the main difference of this study, i.e. added value of this study? It would be better if it is described in comparison of previous US study.

4. What is the future scope of the proposed research, authors have described the limitations in a good way, and I suggest that these can be the future scope of the work.

Author Response

  1. The introduction section introduced some relevant articles. Please explain the results or summarize with effect sizes. 
    Dear Reviewer, thank you for your suggestions. We implemented the indroduction section as you suggested. Please see lines 60-62, 71-74, and 76-78.
  2. Authors selected Eudravigilance. What is the benefit of the Eudravigilance compared to most well-known pharmacovigilance dataset (WHO Vigibase)? Why authors choose it?
    Dear reviewer, we choose to use Eudravigilance, because according to a transparency policy, the data are publicly available through the EMA website (www.adrreports.eu). So, we retrieved data from the open access level of EV database. Instead, to our knowledge, authorization and a longer process are required to access data of the WHO database.
  3. What is the future scope of the proposed research, authors have described the limitations in a good way, and I suggest that these can be the future scope of the work.
    Dear Reviewer, we implement the discussion, as you suggested. Please see lines 426-428

Reviewer 3 Report

Interesting and challenging work. Credible conclusions are particularly important. I think it would be good to clarify a few facts
- were there physically active patients in the group? what type of activity? did it affect the risk of AF?
- how was diabetes treated?
whether and what percentage of patients with obesity?
-please explain the use of ivabradine - has it been used in patients with AF?
- whether drugs that may affect the frequency of AF have been used, for example, beta-blockers, SGLT2 inhibitors, etc.
- has anticoagulant treatment been used? LMWH? DOAC?
- was TDM of DOAC used due to potential interactions?
- has nintedanib been used? if so, were interactions with DOAC taken into account and TDM used?

Author Response

- were there physically active patients in the group? what type of activity? did it affect the risk of AF?
- how was diabetes treated? whether and what percentage of patients with obesity?
Dear reviewer, thank you for your suggestions.  This information is not available in our dataset retrieved from EV. We implemented the discussion section with this limitation in lines 421-423.

-please explain the use of ivabradine - has it been used in patients with AF?
Ivabradine was reported in 5 reports as a concomitant drug. In particular the specific reported therapeutic indication were: generic cardiac disorder (N=1), cardiac failure (N=1), palpitations (N=1), atrial fibrillation (N=1), while in 1 report its indication was not specified.

- whether drugs that may affect the frequency of AF have been used, for example, beta-blockers, SGLT2 inhibitors, etc.
- has anticoagulant treatment been used? LMWH? DOAC?
Dear reviewer, we added this information about the concomitant drugs whose therapeutic indication was atrial fibrillation. Please lines 252-256.

- was TDM of DOAC used due to potential interactions?
Dear Reviewer, considering the dataset type, this information is not available.

- has nintedanib been used? if so, were interactions with DOAC taken into account and TDM used?
Dear reviewer, nintedanib was not reported in any reports.

Round 2

Reviewer 2 Report

All comments have been addressed. Thank you to the authors and editors for considering my opinion on this manuscript.

Reviewer 3 Report

I believe that after clearly defining the limitations of the work as well as extending the manuscript, it can be considered for publication